# Functional social support: A systematic review and standardized comparison of different versions of the DUFSS questionnaire using the EMPRO tool

**Cristina M. Lozano-Hernández**[ORCID][1,2,3,4]*, **Yolanda Pardo Cladellas**[5,6,7], **Mario Gil Conesa**[8], **Olatz Garin**[5,9], **Montserrat Ferrer Forés**[5,6,9], **Isabel del Cura-González**[2,3,10]

**1** Interuniversity Doctoral Program in Epidemiology and Public Health, Rey Juan Carlos University, Alcorcon (Madrid), Spain, **2** Research Unit, Primary Health Care Management Madrid, Madrid, Spain, **3** Research Network on Chronicity, Primary Care and Health Promotion -RICAPPS- (RICORS), Spain, **4** Biosanitary Research and Innovation Foundation of Primary Care (FIIBAP), Spain, **5** Networking Group of the Centre for Biomedical Research, Epidemiology and Public Health (CIBERESP), Instituto de Salud Carlos III, Madrid, Spain, **6** Health Services Research Unit, Institut Mar d'Investigacion Mèdiques (IMIM-Hospital del Mar), Barcelona, Spain, **7** Autonomous University of Barcelona, Spain, **8** Preventive Medicine and Public Health Physician, Castellón General Hospital, Castellón, Valencian Community, Castellón, Spain, **9** Pompeu Fabra University, Barcelona, Spain, **10** Department of Medical Specialties and Public Health. Faculty of Health Sciences Rey Juan Carlos University, Madrid, Spain

* cristinamaria.lozano@salud.madrid.org

## Abstract

### Background

Functional social support is one of the most established predictors of health, and the Duke-UNC Functional Social Support Questionnaire (DUFSS) is one of the most commonly used instruments to measure this parameter. The objective of this study is to systematically review the available evidence on the psychometric and administration characteristics of the different versions of the DUFSS and perform a standardized assessment though to a specifically designed tool.

### Methods

A systematic review was performed in the PubMed/MEDLINE, SCOPUS, WOS and SCIELO databases. All articles that contained information on the development process of the instrument, the psychometric properties and aspects related to its administration were included, without restrictions based on publication date, language, or the version of the questionnaire that was studied. The selection and extraction procedure were carried out by two researchers. The articles finally included were peer-reviewed through a standardised assessment using the Evaluating the Measurement of Patient-Reported Outcomes (EMPRO) tool. PROSPERO registration number: CRD42022342977.

### Results

A total of 54 articles were identified. After eliminating duplicates and screening articles based on the selection criteria, 15 studies that examined the DUFSS questionnaire resulting

**Data Availability Statement:** All relevant data are within the paper and its Supporting Information files. The dataset corresponding to the evaluations

of the different versions of the DUFSS questionnaire using the EMPRO tool has been published in Zenodo. The publication is under DOI: https://doi.org/10.5281/zenodo.8211219.

**Funding:** The main author CMLH received a grant for the translation of this paper from the Foundation for Biosanitary Research and Innovation in Primary Care of the Community of Madrid (FIIBAP). This study was funded by the Fondo de Investigaciones Sanitarias ISCIII (Grant Numbers PI15/00276, PI15/00572, PI15/00996), REDISSEC (Project Numbers RD16/0001/0006, RD16/0001/ 0005 and RD16/0001/0004), and the European Regional Development Fund ("A way to build Europe"). Funders had no role in study design or in the decision to submit the report for publication. The publication of study results was not contingent on the sponsor's approval or censorship of the manuscript.

**Competing interests:** The authors have declared that no competing interests exist.

in 4 different versions: 3 articles obtained the 8-item version; 11 the 11-item version; and a single article obtained two versions, the 14-item version and the 5-item version. At least 60% of them did so in a young adult population, predominantly female and with a medium-low socio-economic level or with characteristics of social vulnerability. The EMPRO evaluation showed that the 11-item version (54.01 total score) was the only one that had been studied on all recommended attributes and had higher total scores than the other versions: 8 items (36.31 total score), 14 items (27.48 total score) and 5 items (23.81 total score). This difference appears in all attributes studied, with the highest scores in "reliability (internal consistency)" and "validity".

## Conclusions

Of the 4 versions identified in the DUFSS questionnaire, the 11-item version was found to be optimal based on the EMPRO standardized tool. Although, a priori, we could prioritise its use in epidemiological studies over the other versions, it should be noted that this version should also be used with caution because there are attributes that have not been studied.

## Background

Research on the relationship between social support and the state of health peaked in the 1970s [1]. Since then, several authors have shown the positive effects of social support on health outcomes [2–4]. Cobb and Cassell [5, 6] argue that the main protective role of social support lies in its moderating effect on life stress. Cohen and Gallant [2, 7] point out that social support impacts on behaviour and the way people manage their health problems and self-care, determining their lifestyles.

There are several definitions of social support across different disciplines [8, 9]. Sociologist P. A. Thoits (1982) defines it as the degree to which a person's basic social needs are satisfied through interaction with others, where basic needs are understood as affiliation, affection, belonging, identity, security and approval [10].

Therefore, there is no consensus regarding the definition of social support [11–13]. According to the conceptual model of Barrón A. [14], social support can be understood from three perspectives: structural, contextual and functional. Structural social support studies the so-called support network, which includes all the contacts of the individual; the size, frequency and density of networks are measured, but the availability of this resource is not measured since the support network is not always a source of available social support [15]. Contextual social support addresses the environment and the circumstances that favour or hinder social support [14]. The functional perspective focuses on the subjective assessment that the person makes regarding their own social support based on its availability and accessibility [16]. The meta-analyses performed by Uchino et al. and Dimatteo et al. show how the study of social support in relation to health is carried out from one or several perspectives according to the authors [17, 18].

Functional social support has been described as a stronger predictor of health than the rest of the perspectives of social support [19] Functional social support has three aspects: 1) emotional support, focused on the closest and most intimate social relationships, sources of care and empathy and composed of two dimensions (the confidential dimension and the affective dimension); 2) instrumental support, also called tangible or material support, because it refers

to practical help in tasks, travel or financial aid; and 3) informational support, referring to support in decision-making or useful advice.

In 1983, Broadhead et al. [20] defined the characteristics of the association between functional social support and health. Based on this work and on the strategic recommendations issued by House and Kahn in 1985 regarding the study of social support, [11] Broadhead developed the Duke-UNC Fuctional Social Support (DUFSS) questionnaire in 1988 [16]. This questionnaire was validated in the USA in the context of primary care and on a general population of mostly young adult women with a medium-high socioeconomic status.

The original version[16] comprised 14-items across four dimensions: "amount of support", "confidential support", "affective support" and "instrumental support". After examining the test-retest reliability, an 11-item version was obtained. This version included the confidential dimension (CF) and the affective dimension (AF) and measured emotional support but omitted measures of the amount of support and instrumental support. The subsequent factor analysis revealed that 3 of the 11 items did not correspond to the resulting dimensions in the other versions, indicating that further research was necessary and thus suggesting that the 8-item version be applied. The final 8-item version included a CF composed of items 3, 4, 5, 6 and 7; and an AF composed of items 1, 2 and 8.

Later (1991) De la Revilla et al. validated the Broadhead questionnaire, by taking its 11-item version, in Spain in the context of primary care and on a general population of mostly young adult women with a low socio-economic status. As a result, they obtained a version with the same number of items (11-items) with a different distribution in its dimensions: CF composed of items 7,8,6,4,1 and 10; and AF composed of items 11,9,2,3 and 5. This questionnaire has been widely used to study functional social support in National Health Surveys, as is the case in Spain [21], and in European surveys, such as the European Health Interview Survey [22]. Therefore, different versions of the DUFSS questionnaire exist, all of which have been validated across very different populations. Thus, it difficult to choose the most appropriate version.

The concept of social support stands out for its subjective character and nature as Patient Reported Outcomes (PRO). According to the recommendations established, a proper validation of a PRO should address certain attributes (FDA, Valderas y Argimon): conceptual and measurement model, reliability, validity, responsiveness and interpretability. In the case of the DUFSS questionnaire, the quality of measurement of social support of each of the different validations is unclear, as no work to date has provided information on this.

The objective of this study was to systematically review the available evidence on the psychometric and administration characteristics of the different versions of the DUFSS questionnaire and perform a standardized assessment though to a specifically designed tool.

## Methods

### Protocol and registration

A systematic review of the literature was carried out, and the results were reported in accordance with the guidelines of the Preferred Reporting Items of Systematic Reviews and Meta-Analyses Protocol (PRISMA). The protocol was registered in PROSPERO under registration number CRD42022342977.

### Eligibility criteria

All articles that contained information on the development of the instrument, the psychometric properties and aspects related to the administration of the Duke-UNC Functional Social Support (DUFSS) questionnaire were included. In order to make the search as sensitive as

possible, there were no restrictions based on publication date, neither on the format (paper or digital), nor on the language of the article or the version or language of the questionnaire used. Regarding the study population, only studies conducted in a population under 18 years of age were excluded, but there were no restrictions based on other population characteristics or settings.

## Information sources

The search was performed on 21/02/2023 in the PubMed/MEDLINE, SCIELO, SCOPUS and WOS databases with the aim of searching a broad swath of databases.

## Search strategy

To develop the search strategy, the different names used for this questionnaire were taken into account. The search strategy used was adapted to each of the databases, in which the terms that appear in Table 1 were included. The reference lists of the included articles were manually searched, and authors were contacted to obtain additional data if necessary. Table 1 shows the terms that have been used to build the search strategy.

## Selection and data collection process

Initially, two reviewers screened the titles according to the inclusion criteria. Then, the same two reviewers did the same for the abstracts. Once the duplicates had been removed and based on the selection criteria, the eligibility of the full articles was assessed. Discrepancies that arose at each of the selection stages were resolved by discussion and consensus between the two researchers, and a third reviewer was consulted when consensus could not be reached between the two previous reviewers.

## Data items

The selected studies were grouped by version type according to the number of items that made up the version resulting from their study. The unit of analysis through EMPRO was each version type of the DUFSS. The following data were extracted: author and year; the version of the questionnaire (number of items and language); the characteristics of the population and country; and the results obtained from the factor analysis carried out, whether exploratory and/or confirmatory: the dimensionality of the questionnaire (unifactorial or bifactorial with the items that make up each dimension). The evaluation and synthesis strategy of the selected articles was included stratifying the articles based on the versions of the DUFSS questionnaire.

**Table 1. Search strategy.**

| |
|---|
| • *"Social support"* |
| **AND** |
| • *"Duke Unc" OR "DUFSS" OR "FSSQ".* |
| **AND** |
| • *Questionnaire\* OR instrument\* OR scale\* OR index\* OR survey\* OR batter\* OR inventor\* OR measur\* OR rating\*.* |
| **AND** |
| • *Valid\*OR Chronbach\* OR "psychometric properties" OR psychometr\* OR Factor Analysis,Statistical[MeSH] OR develop\* OR valid\* OR translat\*.* |

## Synthesis methods

**EMPRO tool.** The EMPRO tool [23] was designed to measure the quality of PRO instruments. This tool demonstrated excellent reliability in terms of internal consistency (Cronbach's alpha = 0.95) and inter-rater concordance (intraclass correlation coefficient: 0.87–0.94). It evaluates quality as a global concept with 39 items across eight attributes (Table 2): "conceptual and measurement model" (concepts and population to be evaluated); " reliability " (to what extent an instrument is free of random errors); " validity " (to what extent an instrument measures what it intends); "sensitivity to changes" (ability to detect changes over time); "interpretability" (assignment of meanings to instrument scores); "burden" (time, effort and other administration and response requirements); "alternative modes of administration" (self-administered or heteroadministered and route of administration); and " cultural and linguistic adaptations". Responses on each item are given on 4-point Likert scale, where 4 is "totally agree", 1 is "totally disagree". Other response options include "no information" and "not applicable". The items answered as "no information" were assigned a score of 1 (lowest possible score) if at least 50% of all items for one attribute were rated; b) items rated as "not applicable" (an option that is only available as an answer for 5 items) were not considered as part of the attribute score.

## Standardized assessment

Each instrument was evaluated by two different experts using the EMPRO tool. Three experts in measuring patient reported outcomes (PROs) composed the review group: two were senior researchers who belonged to the EMPRO tool development working group, and the third was a junior researcher who had been previously trained as an EMPRO evaluator. The pairs of reviewers were composed of a senior and a junior researcher. To minimize the likelihood of bias, the experts were not authors, nor had they participated in the process of development or adaptation of the assigned instrument.

The EMPRO evaluation process consisted of two consecutive rounds. In the first round, each expert independently evaluated the instrument that had been assigned to them from the full-text articles identified. In the second round, each expert received the results of the rating assigned by their review partner. Discrepancies were resolved by discussion or by consulting a third reviewer.

## Analytic strategy

To carry out the analysis of the information, first, the identified studies were stratified according to the resulting item version; second, the published recommendations of the EMPRO tool were followed for the calculation of the scores [23, 24]. For this, at least half of the items that made up each attribute had to be rated from 1 to 4 (responses of "no information" were assigned a score of 1). The mean score was transformed into a scale ranging from 0 (the worst possible score) to 100 (the best possible score). The attributes "reliability" and "load" are composed of two subattributes each: "internal consistency and reproducibility" and "response load and administration load", respectively. For reliability, the highest subscore of its components was chosen to represent the attribute. The calculation of the global score was performed with the average of the first five attributes, since the attribute of load and alternative versions are not metric characteristics but management characteristics. The scores of the attribute of alternative forms were not calculated because there were no validations for different forms of administration (it was self-reported, in some cases via interview), and the scores of the attribute of cultural and linguistic adaptations were not calculated because it was beyond the scope of this study.

**Table 2. Attributes assessed using the evaluating the measurement of patient-reported outcomes (EMPRO) tool.**

| Attribute | Definition | Items included |
|---|---|---|
| Conceptual and measurement model | The rationale for and description of the concept and the populations that a measure is intended to assess and the relationship between these concepts | 1. Concept of measurement stated |
| | | 2. Obtaining and combining items described |
| | | 3. Rationality for dimensionality and scales |
| | | 4. Involvement of target population |
| | | 5. Scale variability described and adequate |
| | | 6. Level of measurement described |
| | | 7. Procedures for deriving scores |
| Reliability | The degree to which an instrument is free from random error | Internal consistency: |
| | | 11. Data collection methods described |
| | | 12. Cronbach's alpha adequate (QA) |
| | | 13. IRT estimates provided |
| | | 14. Testing in different populations |
| | | Reproducibility: |
| | | 15. Data collection methods described |
| | | 16. Test–retest and time interval adequate |
| | | 17. Reproducibility coefficients adequate (QA) |
| | | 18. IRT estimates provided |
| Validity | The degree to which the instrument measures what it purports to measure. | 19. Content validity adequate |
| | | 20. Construct/criterion validity adequate |
| | | 21. Sample composition described |
| | | 22. Prior hypothesis stated (QA) |
| | | 23. Rational for criterion validity |
| | | 24. Tested in different populations |
| Responsiveness | An instrument's ability to detect change over time | 25. Adequacy of methods (QA) |
| | | 26. Description of estimated magnitude of change |
| | | 27. Comparison of stable and unstable groups |
| Interpretability | The degree to which one can assign easily understood meaning to an instrument's quantitative scores. | 28. Rational of external criteria |
| | | 29. Description of interpretation strategies |
| | | 30. How data should be reported stated |
| Burden | The time, effort, and other demands placed on those to whom the instrument is administered (respondent burden) or on those who administer the instrument (administrative burden) | Respondent: |
| | | 31. Skills and time needed |
| | | 32. Impact on respondents |
| | | 33. Not suitable circumstances |
| | | Administrative: |
| | | 34. Resources required |
| | | 35. Time required |
| | | 36. Training and expertise needed |
| | | 37. Burden of score calculation |
| Alternative modes of administration | Alternative modes of administration used for the administration of the instrument | 38. The metric characteristics and use of each alternative mode of administration |
| | | 39. Comparability of alternative modes of administration |

(*Continued*)

**Table 2.** (Continued)

| Attribute | Definition | Items included |
|---|---|---|
| Cultural adaptation | Cultural and linguistic adaptation of the instrument. | 8. Linguistic equivalence (QA) |
| | | 9. Conceptual equivalence |
| | | 10 Differences between the original and the adapted versions |

QA: quality assessed

The overall score of the tool gives each item values from 0 (the worst possible score) to 100 (the best possible score), resulting in an overall average score based on the attributes. The result is considered adequate if it reaches at least 50 points.

## Results

The search strategy (Table 1) yielded 54 studies; 52 studies were obtained from the databases used, and the remaining 2 articles were obtained via manual search. After eliminating duplicates, 30 potentially eligible studies remained. After screening the titles and abstracts, 23 studies remained for full text review. Eight studies were then excluded, including 2 meeting abstracts and 6 studies that examined a different tool that also used the abbreviation DUFFS. Finally, a total of 15 articles were included in this review. The flow diagram is detailed in Fig 1.

Table 3 shows the main characteristics of each of the published validations of the DUFSS questionnaire. Of the 15 studies included, 73.3% of them examined the 11-item version while the rest start from the 8-item and 14-item version in equal parts (13.3% each). The studies that used the 11-item version as a reference were carried out in Spanish-speaking countries [15, 25–32] and in European countries, Italy [33] and Portugal [34], while those that used the 8-item and 14-item versions were carried out in English-speaking countries [16, 35–37], mostly in the USA.

The original version of the questionnaire was designed as a tool to be used in the primary care setting. Thirty-three percent of the localized validations were performed in the field of primary care, and the rest were performed in educational or specific care settings, such as mental health care centres or among pregnant mothers. Regarding the sociodemographic characteristics of the population in which they have been validated, 66.7% have been performed on a predominantly female population; 60% in young adults, whose average age did not exceed 40 years, and 60% in population with medium-low socioeconomic status or who met characteristics of social vulnerability.

From a methodological point of view, only 20% of the included studies used a confirmatory factor analysis to examine the factorial structure of the questionnaire and the dimensions in which its items are grouped. In the exploratory factor analysis, a total of 73.3% of studies reported a two-dimension structure, consisting of the dimensions obtained by Broadhead (confidential and affective). In contrast, 20% of the studies reported a one-dimensional structure, and 6% of studies, all of which examined the modified 11-item version, reported a three-dimensional structure (confidential, affective and instrumental).

Across the 15 studies, the characteristics of 4 different versions of the DUFSS questionnaire were evaluated: the 14-item, 11-item, 8-item and 5-item version. All versions use a Likert scale with 5 response options, except for the 5-item scale, which used both a 5-point Likert scale and a 3-point Likert scale.

Table 4 shows the items that make up each of the versions. The elimination of some items in successive versions has resulted in the reestablishment of the numbering values. Therefore, the 8-item and 5-item versions do not share any numbering with the original version. A study performed on the 11-item version revealed that 2 new items were included but were not part of the original 14-item version.

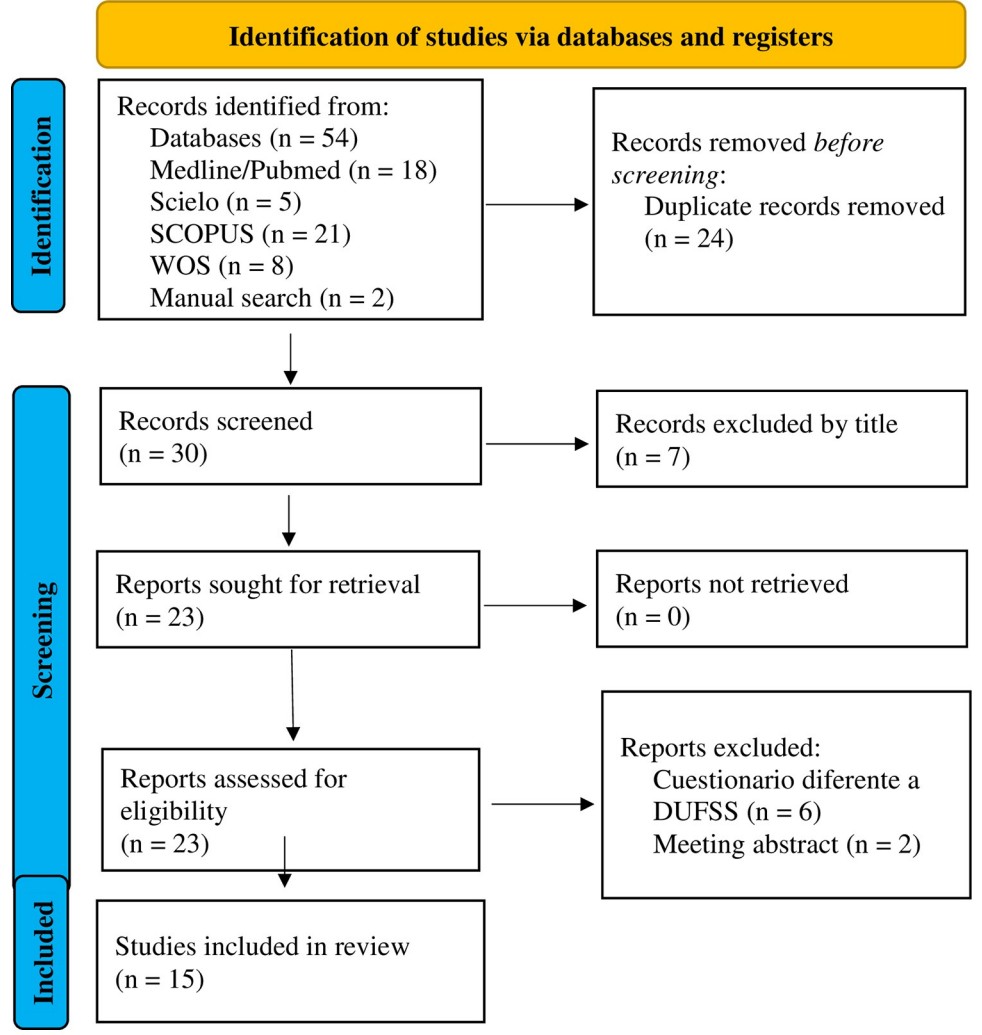

**Fig 1. PRISMA flow chart—systematic literature search.**

The results of the EMPRO evaluation about psychometric values of each of each version are shown in Table 5. The highest score was obtained by the 11-item version (54.01 points), followed by the 8-item version (36.31 points), the 14-item version (27.48 points) and the 5-item version (23.81 points).

### Conceptual and measurement model

The 11-item version obtained the highest score for this attribute (54.76 points), while the rest of the versions obtained a score of 35.71 points. The aspects that were least addressed by the included studies were the description of the measurement scale (including its scores) and the participation of the sample in a previous pilot.

### Reliability

Only studies of the 11-item version evaluated both aspects; the other studies just examined the data collection method and calculated Cronbach's alpha coefficient without assessing internal consistency or reproducibility. The highest internal consistency score was obtained by the

**Table 3. Main characteristics of the DUFSS validations.**

| Resulting version | Author/year | Language version | Version used | Setting & Country | Sample | Results |
|---|---|---|---|---|---|---|
| *8-item version* | Broadhead et al. [**1988**] [16] | English | 14-item | Primary Care General Population USA | n = 401 Women: 78% Age: 35.7 (± *) Medium-high socioeconomic status | **AFE** CF: 3, 4, 5, 6 and 7 AF: 1, 2 and 8 |
| | Kathy B Isaacs et al. [**2011**] [35] | English | 8-items | Specialized centre for pregnant women USA | n = 186 Women: 100% - Low socioeconomic status | **AFE** Unifactorial |
| | H. M. Epino et al. [**2012**] [36] | English | 8-item | Rural Primary Care HIV-positive Rwanda | n = 603 Women: 62% Age: 38 ± 10 Low socioeconomic status | **AFE** Unifactorial |
| *11-item version* | De la Revilla et al. [**1991**] [25] | Spanish | 11-item | Primary Care General Population Spain | n = 139 Women: 82% Age: 46 ± 17.6 Low socioeconomic status | **AFE** CF: 7,8,6,4,1 and 10 AF: 11,9,2,3 and 5 |
| | Bellón S. JA. et al. [**1996** [15] | Spanish | 11-item | Primary Care General Population Spain | n = 656 Women: 72% Age: 50.6 ± 18.9 Low socioeconomic status | **AFE** CF: 1, 2, 6, 7,8,9 and 10 AF: 3, 4, 5 and 11 |
| | Alvarado BE. et al. [**2005**] [26] | Spanish | 11-item | Municipal Population Register Mothers of children between 6–18 m Colombia | n = 193 Women: 100% - | **AFE** CF: 4,5,10 and 11 AF: 6,7,8 |
| | Piña L. A. et al. [**2007**] [27] | Spanish | 11-item | Rural specialized centre for HIV-positive individuals Mexico | n = 67 Women: 32.40% Age: 36.4 ± 10.6 Low socioeconomic status | **AFE** Unifactorial |
| | Ayala A. et al. [2012] [28] | Spanish | 11-item | Municipal Population Register Noninstitutionalized seniors Spain | n = 1012 Women: 56.30% Age: 72.1 ± 7.8 - | **AFE** CF: 7, 8, 6, 5, 4, 11 and 10; AF: 2, 1, 9 and 3 |
| | Cuellar-Flores L. et al. [2012] [29] | Spanish | 11-item | Primary Care Caregivers Spain | n = 128 Women: 85.90% Age: 54.9 ± 15.1 - | **AFE** CF: 2, 6, 7, 8, 9, 10 and 11; AF: 1, 3, 4 and 5 |
| | Mas-Exposito L. et al. [**2013**] [30] | Spanish | 11-item | Specialized centre for people with mental illness Spain | n = 241 Women: 32.40% Age: 41.7 ± 11.6 Low socioeconomic status | **AFE** CF: 4, 6,7,8, 10 and 11; AF: 1, 2, 9, 3 and 5 |
| | Rivas Diez [2013] [31] | Spanish | 11-item | Educational centres Chile | n = 371 Women: 100% Age: 37.6 ± 13.1 Medium-high socioeconomic status | **AFE/AFC** CF: 4, 5, 6, 7, 8, 10 and 11 AF: 1, 2, 3 and 9 |
| | | | | Specialized centre for victims of abuse Chile | n = 97 Women: 100% Age: 41.9 ± 10 Medium-low socioeconomic status | **AFE/AFC** CF: 3, 5, 6, 7, 8, 9 and 10 AF: 1, 2, 4 and 11 |
| | Caycho R. T. et al. [**2014**] [33] | Italian | 11-item | Specialized Centre for Peruvian Migrants Italy | n = 150 Women: 58% Age: 34.6 ± 10.3 Medium-low socioeconomic status | **AFC** CF: 1,4,6,7,8 y10; AF: 2, 3,5, 9 and 11 |
| | Mónica Aguilar-Sizer et al. [2021] [32] | Spanish | 11-item | Educational centre for the general population Ecuador | n = 535 Women: 75.5% Age: 22± 5 | **AFE** CF: 6, 7 and 8 AF: 1, 2, 3, 4, 5, 9, 10 and 11 |
| | Martins, S. et al. [**2022**] [34] | Portuguese | 11-item | Mother and fathers of young children educational centres Portugal | n = 1.058 Women: 90.5% Age: 35.7± 5.2 Medium-high socioeconomic status | **AFC** CF: 1, 6, 7,8,9 and 10 AF: 3, 4 and 5 IF: 2, 11, 12 and 13 |

*(Continued)*

**Table 3.** (Continued)

| Resulting version | Author/year | Language version | Version used | Setting & Country | Sample | Results |
|---|---|---|---|---|---|---|
| *14-item version* | **Rebecca Saracino et al. [2014]** [37] | English | 14-item | Specialized centres for patients with incurable and advanced diseases (AIDS and cancer) USA | n = 253 Women: 69.6% Age: 58.2± 11 - | **AFE** Unifactorial |
| *5-item/ 3-response version* | | | | | | **AFE** Unifactorial |

\* Data not available. Factor analysis performed: Confirmatory Factor Analysis (CFA), Exploratory Factor Analysis (EFA)

CF (Confidential Factor), AF (Affective Factor) and FI (Instrumental Factor)

11-item version (70.83 points), because Cronbach's alpha in all studies was higher than 0.9. In addition, most of the internal consistency quality criteria, with the exception of the IRT criterion, also had high scores (three or four crosses). Although most studies reported an adequate Cronbach's alpha coefficient ($\geq 0.70$), they did not comprehensively measure the reliability of the instrument.

## Validity

Only one study, from the 11-item version, used the EFA and CFA to examine content validity. The versions with the highest scores for this attribute were the 11-item version (66.67 points) and the 8-item version (62.50 points). In the included studies, differences were been observed between the dimensions of the DUFSS both in the distribution of its items and in the number of dimensions included in the tool. The relationship with previous hypotheses and related

**Table 4. Different versions of the DUFSS questionnaire by number of items.**

| 14-item version* | 11-item version* | 11-item version modified* | 8-item version* | 5-item version** |
|---|---|---|---|---|
| **Item 1:** *visits with friends and relatives.* | Item 1 | Item 1 | Item 5 | Item 5 |
| **Item 2:** *help around the house.* | Item 2 | Item 2 | Item 6 | Item 6 |
| **Item 3:** *help with money in an emergency.* | Item 4 | Item 4. | Item 8 | Item 9 |
| **Item 4:** *praise for a good job.* | Item 5 | Item 5 | Item 9 | Item 12 |
| **Item 5:** *people who care what happens to me.* | Item 6 | Item 6 | Item 10 | *help when I need transportation.* |
| **Item 6:** *love and affection.* | Item 8 | Item 8 | Item 11 | |
| **Item 7:** *telephone calls from people I Know.* | Item 9 | Item 11 | Item 12 | |
| **Item 8:** *chances to talk to someone about problems at work or with my housework.* | Item 10 | Item 9 | Item 14 | |
| **Item 9:** *chances to talk to someone I trust about my personal and family problems.* | Item 11 | Item 12 | | |
| **Item 10:** *chances to talk about money matters.* | Item 12 | Item 10 | | |
| **Item 11:** *invitations to go out and do things with other people.* | Item 14 | Item 14 | | |
| **Item 12:** *I get useful advice about important things in life.* | | *help with transportation and move* | | |
| **Item 13:** *help when I need transportation.* | | *help with my children's care* | | |
| **Item 14:** *help when I'm sick in bed.* | | | | |

*5-response Likert-type

**3-response Likert-type

**Table 5. EMPRO results: psychometric values of each version of the DUFSS questionnaire.**

| Attributes | 14-Items | 11-Items | 8-Items | 5-Items |
|---|---|---|---|---|
| **Conceptual and measurement model** | **35.71** | **54.76** | **35.71** | **35.71** |
| 1. Concept of measurement stated | +++I | +++I | +++I | +++I |
| 2. Obtaining and combining items described | ++ | +++ | ++ | ++ |
| 3. Rationality for dimensionality and scales | +++ | +++ | ++I | +++ |
| 4. Involvement of target population | + | ++ | + | + |
| 5. Scale variability described and adequate | + | +++ | +I | + |
| 6. Level of measurement described | ++ | ++ | ++ | ++ |
| 7. Procedures for deriving scores | ++ | ++ | ++ | ++ |
| **Reliability** | **41.66** | **70.83** | **44.44** | **25** |
| Internal consistency | 41.66 | 70.83 | 44.44 | 25 |
| 8. Data collection methods described | +++ | +++I | ++I | ++ |
| 9. Cronbach's alpha adequate | ++++ | +++ | +++I | +++ |
| 10. IRT estimates provided | NI | +++ | NI | NI |
| 11. Testing in different populations | NI | +++ | NI | NI |
| Reproducibility | - | 45.83 | - | - |
| 12. Data collection methods described | NI | +++I | NI | NI |
| 13. Test–retest and time interval adequate | NI | ++ | NI | NI |
| 14. Reproducibility coefficients adequate | + | +++ | NI | NI |
| 15. IRT* estimates provided | NI | NI | NI | NI |
| **Validity** | **26.67** | **66.67** | **62.50** | **25** |
| 16. Content validity adequate | + | ++ | +I | + |
| 17. Construct/criterion validity adequate | ++ | +++I | ++I | ++ |
| 18. Sample composition described | +++ | +++I | ++++ | +++I |
| 19. Prior hypothesis stated | ++ | +++ | +++I | NI |
| 20. Rational for criterion validity | NA | NA | NA | ++ |
| 21. Tested in different populations | + | +++ | NA | + |
| **Responsiveness** | **-** | **33.33** | **-** | **-** |
| 22. Adequacy of methods | NI | +++ | NI | NI |
| 23. Description of estimated magnitude of change | NI | ++ | NI | NI |
| 24. Comparison of stable and unstable groups | NI | NI | NI | NI |
| **Interpretability** | **33.33** | **44.44** | **38.89** | **33.33** |
| 25. Rational of external criteria | ++ | +++ | ++I | ++ |
| 26. Description of interpretation strategies | NI | ++ | NI | NI |
| 27. How data should be reported stated | +++ | ++ | +++ | +++ |
| **Overall score** | **27.48** | **54.01** | **36.31** | **23.81** |
| **Burden** | | | | |
| **Burden: respondent** | **33.33** | **66.67** | **38.89** | **33.33** |
| 28. Skills and time needed | +I | +++I | ++ | +I |
| 29. Impact on respondents | +I | ++I | ++ | +I |
| 30. Not suitable circumstances | +++ | +++ | ++I | +++ |
| **Burden: administrative** | **33.33** | **50.00** | **66.67** | **66.67** |
| 31. Resources required | NI | +++ | ++I | +++ |
| 32. Time required | NA | NA | NA | NA |
| 33. Training and expertise needed | NA | NA | NA | NA |
| 34. Burden of score calculation | +++ | ++ | +++I | +++ |

Ítems Score: + 1 point, I ½ point, NI no information, NA. not applicable.

*IRT (Item Response Theory)

variables (convergent validity) was analysed in the 14-, 11- and 8-item versions, which improved their scores on the validity attribute.

## Responsiveness

Regarding the sensitivity to change, only the 11-item version detects the sensitivity to change compared with of other validated scales (33.33 points); however, the 11-item version does not offer comparative results between groups (longitudinal validity), because most of the included studies used a cross-sectional design. In the rest of the versions, not enough information was found to calculate scores.

## Interpretability

The interpretability of the questionnaire was similar across all versions (with a range of 33.33 to 44.44 points); the 11-item version had the highest score, followed by the 8-item version. None of the 4 versions studied herein offer sufficient information on the measurement and interpretation strategies of the DUFSS questionnaire. Only Bellón et al. (11-item version) provided information by stating that the 15th percentile (score ≤32) of their sample was the cut-off point for differentiating "good" from "low" social support.

## Burden

Regarding response load and administration load, both attributes had scores ranging from 33.33 to 66.67. The 11-item version obtained a higher score for response load, whereas the 8- and 5-item versions obtained the highest scores for administration load.

# Discussion

## Main results

This systematic review has identified 15 studies on the validation of the DUFSS that have obtained different versions of the questionnaire according to the number of resulting items: 14-items, 11-items, 8-items and 5-items. Validations carried out in Spanish-speaking countries predominate in which, as in Italy and Portugal, the 11-item version validated by De La Revilla et al. was used, while in English-speaking countries, mostly in the USA, the 8-item and 14-item version validated by Broadhead et al. was used for their study (citation). Despite the fact that the origin of the questionnaire is American and that more than half of the studies were carried out in countries with different languages (Spanish, Portuguese, Italian and English) and in very specific populations (young adults, women and with a medium-low socio-economic level or with characteristics of social vulnerability), none of the studies analysed describe the process used to translate and culturally adapt the instrument to their study populations. This aspect could call into question the content validity of the different versions of the questionnaire in relation to the original. Comparing this reality with that of similar studies, we found that the 12-item Multidimensional Scale of Perceived Social Support (MSPSS) [28] was assessed for the psychometric properties of its existing translations using the COSMIN tool [29] and concluded that the translated versions provided little evidence for content validity. This absence may explain the differences in the distribution of its items as a result of the EFA and ACE (Table 4).

The quality assessment of the DUFSS questionnaire through the EMPRO tool showed that the 11-item version is the only one that has been studied through all the recommended attributes and the one that has obtained the highest score in its evaluation. This version scored higher than the rest in all the attributes studied, with the best scores for "reliability (internal consistency)" and "validity"; its lowest score was for the attribute "responsiveness", where the

rest of the versions had no information (N/A). This result is probably due to the fact that the 11-item version has been studied by 73.3% of the studies analysed, while the other versions have less validations. The 8-item version obtained its best score in the attribute "validity", and attributes such as "responsiveness" and "reliability (reproducibility)"; the 14-item version obtained its highest score in the section "reliability (internal consistency)", despite the lack of information in "reliability (reproducibility)"; the 8-item version did not present information in the attributes "responsiveness" and "reliability (reproducibility)", hence the resulting low score. The complex nature of PRO instruments often raises important questions about how to interpret and communicate the results in a way that is not misleading, so it is essential that the validation of the instrument makes clear how the results are to be interpreted. None of the 4 versions studied herein offer sufficient information on the measurement and interpretation strategies of the DUFSS questionnaire. One of the most frequent strategies for interpreting PROs is to calculate percentiles from population values [38]. This method is only valid if the data come from a representative sample of the general population; however, this was not the case in the included studies. Bellón et al. found that the 15th percentile was the cut-off point to differentiate "good" from "low" social support. This percentile corresponded to a score ≤32, a score that later Fernández et al. [39] and Ruiz et al. [40] applied in their studies. On the other hand, authors such as Slade et al. [41] and Harley et al. [42] and the National Health Survey of Spain [43] expressed the results of social support in the form of a quantitative variable. The latter option seems to be the most appropriate approach, since it does not require percentiles to be extracted from the general population. However, the most appropriate approach would be to use population values or norms, reference samples, or standardized mean responses.

## Limitations and strengths

This work constitutes the first review of the social support construct using a standardized methodology, the EMPRO tool, for the evaluation of one of the most well-known instruments.

A limitation of this work is that the systematic reviews depend on the information retrieved through the search strategy, so it is possible that we have not identified all the articles published on the questionnaire to be studied. However, given that social support is a construct of interest in multiple disciplines and that some validations might be indexed in non-health science databases, in addition to including Pubmed, we also searched multidisciplinary databases such as SCOPUS and WOS. This aspect added to the delicate search strategy designed, the additional manual search of references, as well as the double independent review process followed, may have minimised this problem [16].

The most important aspect to note as a limitation is that the results shown here do not suggest a single, unequivocal ranking or recommendation, as the EMPRO assessment includes several attributes and they have different relevance according to the purpose of the application of the questionnaire. For example, if the purpose is monitoring patients, responsiveness is the key attribute. Although this tool has shown excellent reliability in terms of internal consistency and inter-rater concordance the EMPRO scores depend on the quantity and quality of information provided for the assessment. Thus, information on psychometric properties that are not available, they are not taken into account and therefore penalise the assessment. On the other hand, EMPRO offers the possibility to answer with "no information" and if the half of items has not information, the attribute score is not calculated. In this regard, it is important to highlight that the work carried out by Ayala et al. [28] on the 11-item version addresses most of the attributes proposed by EMPRO, which increases the score of this version.

EMPRO ratings may be biased by the individual expertise of the evaluators, although the double and independent review conducted, as well as a comprehensive description of each

item, may have attenuated this concern. Studies on metric properties from different country versions were considered in our EMPRO evaluation. Although these country versions can add noise in one sense, they also provide valuable information about the generalizability of the psychometric data to these measures.

The coexistence of different nomenclatures has been a challenge in the process of reviewing and selecting articles. The name of the questionnaire has been modified since the original author called it the Duke-UNC Functional Support Questionnaire (DUFSS). Subsequently, various researchers have modified the original acronym, thus yielding a variety of names: DUKE-UNC, DUKE-UNC-11 DUKE-UNK, FSSQ and DUFSSQ. This increases the confusion among researchers who want to use some version of the DUFSS; additionally, these alternate acronyms may be similar to different tools that measure separate concepts, thus creating the potential for more confusion–e.g., the Duke Social Support Index (DSSI) [44] or the Duke-UNC Health Profile (DUHP) [45]. In order to ensure that all validation articles on the DUFSS were located, we included the possible names it may acquire in the search strategy (this part is explained in the methodology); and once we had the search results, we went on to read the full text of any paper that might raise doubts, in order to ensure that it was the right questionnaire.

## Applicability

This paper can shed light on the study of social support as a PRO in different domains and help to unravel the current complexity that exists around this questionnaire.

To know the different versions of the DUFSS questionnaire and providing relevant information about each one will allow researchers who wish to study this subject to choose the version that best suits their interests and to be aware of the evaluation of its quality. In addition, improving knowledge about this PRO will allow progress to be made and give greater strength to the work in the field of epidemiology and public health on person-centred care.

Furthermore, in the educational field, this study has two applications: on the one hand, to train specific tools in the study of PROs; on the other hand, it focuses on the importance of consulting the original sources and investigating the work that has been done previously on the research question.

## Conclusions

There are 4 versions of the EMPRO questionnaire with different numbers of items: 14, 11, 8 and 5 items. All of them have been validated in very specific populations and not in the general population.

Among the 4 versions the DUFSS questionnaire, the 11-item version has been the most studied, especially in Spanish-speaking countries. This version scored higher than the others because it was the version with the largest number of studies and therefore more likely to address all the attributes taken into account by the EMPRO tool.

In order to be able to state with certainty that the 11-item version is more appropriate than the other versions, more studies are needed to evaluate each of the other versions. Although, a priori, we could prioritise its use in epidemiological studies over the other versions, it should be noted that this version should also be used with caution because there are attributes that have not been studied.

All versions of the DUFSS questionnaire should be used with caution, since many of the attributes studied have not shown sufficient quality in any of the versions analysed herein. It is necessary to conduct future studies on the DUFSS questionnaire to evaluate aspects such as its reproducibility and to perform complete factor analysis in the general population.

## Supporting information

**S1 Checklist. PRISMA 2020 checklist.**
(DOCX)

**S1 Dataset.**
(XLSX)

## Acknowledgments

We thank JA López-Rodríguez for his help in reviewing this work.

## Author Contributions

**Conceptualization:** Cristina M. Lozano-Hernández, Yolanda Pardo Cladellas, Isabel del Cura-González.

**Data curation:** Cristina M. Lozano-Hernández, Yolanda Pardo Cladellas, Mario Gil Conesa, Olatz Garin, Isabel del Cura-González.

**Formal analysis:** Cristina M. Lozano-Hernández, Yolanda Pardo Cladellas, Mario Gil Conesa, Olatz Garin, Isabel del Cura-González.

**Funding acquisition:** Cristina M. Lozano-Hernández, Isabel del Cura-González.

**Investigation:** Cristina M. Lozano-Hernández, Isabel del Cura-González.

**Methodology:** Cristina M. Lozano-Hernández, Yolanda Pardo Cladellas, Mario Gil Conesa, Olatz Garin, Montserrat Ferrer Forés, Isabel del Cura-González.

**Resources:** Cristina M. Lozano-Hernández.

**Supervision:** Montserrat Ferrer Forés, Isabel del Cura-González.

**Validation:** Cristina M. Lozano-Hernández, Yolanda Pardo Cladellas, Isabel del Cura-González.

**Visualization:** Cristina M. Lozano-Hernández.

**Writing – original draft:** Cristina M. Lozano-Hernández.

**Writing – review & editing:** Cristina M. Lozano-Hernández, Yolanda Pardo Cladellas, Montserrat Ferrer Forés, Isabel del Cura-González.

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
