## [Decision Letter · Decision Letter 0]

10 Feb 2023

PONE-D-22-28641Functional Social Support: a systematic review and standardized comparison of different versions of the DUFSS questionnaire using the EMPRO tool.PLOS ONE

Dear Dr. Lozano-Hernández,

Thank you for submitting your manuscript to PLOS ONE. After careful consideration, we feel that it has merit but does not fully meet PLOS ONE’s publication criteria as it currently stands. Therefore, we invite you to submit a revised version of the manuscript that addresses the points raised during the review process. Please submit your revised manuscript by Mar 27 2023 11:59PM. If you will need more time than this to complete your revisions, please reply to this message or contact the journal office at plosone@plos.org. Please include the following items when submitting your revised manuscript:A rebuttal letter that responds to each point raised by the academic editor and reviewer(s). You should upload this letter as a separate file labeled 'Response to Reviewers'.A marked-up copy of your manuscript that highlights changes made to the original version. You should upload this as a separate file labeled 'Revised Manuscript with Track Changes'.An unmarked version of your revised paper without tracked changes. You should upload this as a separate file labeled 'Manuscript'.

We look forward to receiving your revised manuscript.

Kind regards,

Gian Mauro Manzoni, Ph.D., Psy.D.

Academic Editor

PLOS ONE

Journal Requirements:

2. Please note that in order to use the direct billing option the corresponding author must be affiliated with the chosen institute. Please either amend your manuscript to change the affiliation or corresponding author, or email us at plosone@plos.org with a request to remove this option.

Additional Editor Comments:

#Reviewer 1:

The manuscript is very interesting. I have some concerns - but fully addressables.

- Considering the topic of your research, the introduction is quite short and should be improved.

- The EMPRO tool is very intersting and i think that should be describe with major details - allowing other researchers to use the same procedure for their studies (using supplementary material, maybe?).

- Considering that the research of the studies was made on 4th April - and now it's december - I suggest to made another research in order to not exclude new-published studies

- I suggest the Authors to enlarge the limitations and stregths section

#Reviewer 2:

This article provides a thorough analysis and evaluation to compare psychometric performances of different DUFFS versions. As such it provides groundings for researches to justify taking one version over another. Overall, the article is well structured and consistent. Having a summary table reporting psychometrics for each version would help readers interprete results beyond the EMPRO tool with which most readers are not familiar.

Abstract

1. Consider adding more details in the methods section such as the search terms, how many ran the selection and extraction procedure, what data was extracted and what the inclusion criteria where (language restriction, type of publication, etc.).

2. The search method identified 54 articles of which 15 were retained. The actual statement can be confusing as it suggests 54 were retained and only 15 analysed.

3. In the results section, please provide more details on what explains the score difference and in what way the DUFFS-11 was better than the others. Furthermore, provide some indication on compatibility between the studies in the sense of what population was tested and what underlying conditions were present.

4. Consider revising the conclusion as recommendation for choosing one of the versions in an epidemiological study.

Background

5. Consider adding a small paragraph on different expected psychometric measures and in what way they can inform on DUFFS performances.

6. In the wording of the study objectives, it is unclear what is meant by “a standardised way” please consider specifying in what way.

Methods

7. Line 95 – Please give more details on what articles were to be eligible. Are they accepted, published online, in paper, do they have to be peer reviewed? Are there any language restrictions? Are all language versions of the DUFFS to be included?

8. Line 98 – What I semant by “underage population”? Under the age of what?

9. Please consider moving lines 103-107 to the next section “search strategy”.

10. Discrepencies in what? Please reword to make clear if the selection process was done based on titles, than on abstracts than on full text and if discrepancies in choices for selection were treated during each step.

11. Line 116 – Please provide further details on what is meant by “main results”. It is useful to understand what psychometric measures were of interest and how they were extracted.

12. For compatibility between studies, were some measures recalculated based on the information made available?

13. Line 122 – Consider rewording the section heading to “EMPRO tool” and then “Experts”. Please provide further information about the EMPRO tool including its available psychometrics. What does this instrument measure and how valid is this measurement?

14. How was study quality assessed for the included articles?

Results

15. Consider providing more detailed results of extracted psychometrics from the studies in the result section for Conceptual and measurement model, reliability, validity, responsiveness, and interpretability as reported in the discussion.

15b. Adding a table with psychometric values for each version would help.

Discussion

16. Please consider synthesising the results in a paragraph, then to identify limitation of the actual study, and then discuss the results compared to other similar studies on similar instruments, then discuss the practical implications and then conclude.

17. Lines 293-301, please provide deeper insights on cultural and language differences between versions of a questionnaire. To what extend can we assume we are measuring the same thing? What does the literature say about this and in what way could the results between different versions be explained by cultural differences rather than version differences? Do the different versions have similar psychometric values between English and Spanish (to be reported in the results section)?

18. The main limitation is the reliance on the EMPRO tool. It is unclear whether a score can be affected only by good-bad psychometric performances or if it can be affected by the fact some psychometric measures were not made available by research. It is also unclear why each criteria has similar weight for the general score. In what way does that relate to the test value. A test that performed very badly in reliability will remain bad even if it performs well for other psychometric characteristics. Another limitation is combining psychometrics of tests run in different languages. Another limitation is that the version are compared only on psychometric performance but other advantages might exist on some versions over others such as time needed to answer questions, understandability, etc.

19. Lines 311-319; This does not seem to be a limitation as it does not seem to affect the results of this study. Misclassification of studies as been the DUFFS would be a limitation. What measures were taken to make sure that the retained article truly tested the DUFFS?

20. Please consider adding a section on practical implication for research, education and public health.

Conclusion

21. In what way can the 5-item version compete as it only was evaluated by two studies. How can authors justify this classification based on a score that could change in time depending of future available studies on other versions. How can we go beyond saying that with the actual available evidence, the DUFFS that shows the best performances is the 11-item version.

Figures

22. Some text within the boxes have not been translated from Spanish.

Tables and supplement data

23. Extracted psychometric values for each version is not made available.

Reviewers' comments:

Reviewer's Responses to Questions

**Comments to the Author**

1. Is the manuscript technically sound, and do the data support the conclusions?

Reviewer #1: Yes

Reviewer #2: Partly

2. Has the statistical analysis been performed appropriately and rigorously? 

Reviewer #1: Yes

Reviewer #2: N/A

3. Have the authors made all data underlying the findings in their manuscript fully available?

Reviewer #1: Yes

Reviewer #2: No

4. Is the manuscript presented in an intelligible fashion and written in standard English?

Reviewer #1: Yes

Reviewer #2: Yes

5. Review Comments to the Author

Reviewer #1: The manuscript is very interesting. I have some concerns - but fully addressables.

- Considering the topic of your research, the introduction is quite short and should be improved.

- The EMPRO tool is very intersting and i think that should be describe with major details - allowing other researchers to use the same procedure for their studies (using supplementary material, maybe?).

- Considering that the research of the studies was made on 4th April - and now it's december - I suggest to made another research in order to not exclude new-published studies

- I suggest the Authors to enlarge the limitations and stregths section

Reviewer #2: This article provides a thorough analysis and evaluation to compare psychometric performances of different DUFFS versions. As such it provides groundings for researches to justify taking one version over another. Overall, the article is well structured and consistent. Having a summary table reporting psychometrics for each version would help readers interprete results beyond the EMPRO tool with which most readers are not familiar.

Abstract

1. Consider adding more details in the methods section such as the search terms, how many ran the selection and extraction procedure, what data was extracted and what the inclusion criteria where (language restriction, type of publication, etc.).

2. The search method identified 54 articles of which 15 were retained. The actual statement can be confusing as it suggests 54 were retained and only 15 analysed.

3. In the results section, please provide more details on what explains the score difference and in what way the DUFFS-11 was better than the others. Furthermore, provide some indication on compatibility between the studies in the sense of what population was tested and what underlying conditions were present.

4. Consider revising the conclusion as recommendation for choosing one of the versions in an epidemiological study.

Background

5. Consider adding a small paragraph on different expected psychometric measures and in what way they can inform on DUFFS performances.

6. In the wording of the study objectives, it is unclear what is meant by “a standardised way” please consider specifying in what way.

Methods

7. Line 95 – Please give more details on what articles were to be eligible. Are they accepted, published online, in paper, do they have to be peer reviewed? Are there any language restrictions? Are all language versions of the DUFFS to be included?

8. Line 98 – What I semant by “underage population”? Under the age of what?

9. Please consider moving lines 103-107 to the next section “search strategy”.

10. Discrepencies in what? Please reword to make clear if the selection process was done based on titles, than on abstracts than on full text and if discrepancies in choices for selection were treated during each step.

11. Line 116 – Please provide further details on what is meant by “main results”. It is useful to understand what psychometric measures were of interest and how they were extracted.

12. For compatibility between studies, were some measures recalculated based on the information made available?

13. Line 122 – Consider rewording the section heading to “EMPRO tool” and then “Experts”. Please provide further information about the EMPRO tool including its available psychometrics. What does this instrument measure and how valid is this measurement?

14. How was study quality assessed for the included articles?

Results

15. Consider providing more detailed results of extracted psychometrics from the studies in the result section for Conceptual and measurement model, reliability, validity, responsiveness, and interpretability as reported in the discussion.

15b. Adding a table with psychometric values for each version would help.

Discussion

16. Please consider synthesising the results in a paragraph, then to identify limitation of the actual study, and then discuss the results compared to other similar studies on similar instruments, then discuss the practical implications and then conclude.

17. Lines 293-301, please provide deeper insights on cultural and language differences between versions of a questionnaire. To what extend can we assume we are measuring the same thing? What does the literature say about this and in what way could the results between different versions be explained by cultural differences rather than version differences? Do the different versions have similar psychometric values between English and Spanish (to be reported in the results section)?

18. The main limitation is the reliance on the EMPRO tool. It is unclear whether a score can be affected only by good-bad psychometric performances or if it can be affected by the fact some psychometric measures were not made available by research. It is also unclear why each criteria has similar weight for the general score. In what way does that relate to the test value. A test that performed very badly in reliability will remain bad even if it performs well for other psychometric characteristics. Another limitation is combining psychometrics of tests run in different languages. Another limitation is that the version are compared only on psychometric performance but other advantages might exist on some versions over others such as time needed to answer questions, understandability, etc.

19. Lines 311-319; This does not seem to be a limitation as it does not seem to affect the results of this study. Misclassification of studies as been the DUFFS would be a limitation. What measures were taken to make sure that the retained article truly tested the DUFFS?

20. Please consider adding a section on practical implication for research, education and public health.

Conclusion

21. In what way can the 5-item version compete as it only was evaluated by two studies. How can authors justify this classification based on a score that could change in time depending of future available studies on other versions. How can we go beyond saying that with the actual available evidence, the DUFFS that shows the best performances is the 11-item version.

Figures

22. Some text within the boxes have not been translated from Spanish.

Tables and supplement data

23. Extracted psychometric values for each version is not made available.

6. PLOS authors have the option to publish the peer review history of their article (what does this mean?). If published, this will include your full peer review and any attached files.

Reviewer #1: No

Reviewer #2: **Yes: **Paul Vaucher

---

## [Author Response · Author response to Decision Letter 0]

15 Apr 2023

Dear reviewers:

We greatly appreciate the comments and submissions made on this paper. We certainly feel that the quality of the document has been greatly enhanced.

Below are the details of each of the responses requested.

#Reviewer 1:

The manuscript is very interesting. I have some concerns - but fully addressables.

1. Considering the topic of your research, the introduction is quite short and should be improved.

We appreciate the suggestion. We have added information in the introduction that helps to better understand the concept of social support, its relationship to health and aspects of its psychometric measurement.

2. The EMPRO tool is very intersting and i think that should be describe with major details - allowing other researchers to use the same procedure for their studies (using supplementary material, maybe?).

Thank you very much for the proposal. We have prepared a table on the attributes studied by EMPRO. It details each attribute with its name, definition, the number of items it is composed of, and with examples of scoring.

We have included it in the file "Tables and Figures" under the name "Table 2. Attributes assessed using the Evaluating the Measurement of Patient-Reported Outcomes (EMPRO) tool".

3. Considering that the research of the studies was made on 4th April - and now it's December. I suggest to made another research in order to not exclude new-published studies

The search for this systematic review was conducted on 4 April. Following your recommendation, we have included the search again and found that there are no new articles on this topic.

4. I suggest the Authors to enlarge the limitations and stregths section.

Thank you for your suggestion. We have followed your instructions and have enlarged different aspects of the limitations and strengths of this work. 

#Reviewer 2:

This article provides a thorough analysis and evaluation to compare psychometric performances of different DUFFS versions. As such it provides groundings for researches to justify taking one version over another. Overall, the article is well structured and consistent. Having a summary table reporting psychometrics for each version would help readers interprete results beyond the EMPRO tool with which most readers are not familiar.

Abstract:

1. Consider adding more details in the methods section such as the search terms, how many ran the selection and extraction procedure, what data was extracted and what the inclusion criteria where (language restriction, type of publication, etc.).

Following your recommendations, we have partially expanded the information in this section. But taking into consideration the word limit established by the journal's rules for the abstract, the rest of the details have been expanded in detail in the body of the paper. 

2. The search method identified 54 articles of which 15 were retained. The actual statement can be confusing as it suggests 54 were retained and only 15 analysed.

Thank you for notifying us this mistake. This may be due to a translation error. We have replaced the term "retrieved" with "identified".

3. In the results section, please provide more details on what explains the score difference and in what way the DUFFS-11 was better than the others. Furthermore, provide some indication on compatibility between the studies in the sense of what population was tested and what underlying conditions were present.

As requested, we have provided more detailed information in the results section of the summary. 

4. Consider revising the conclusion as recommendation for choosing one of the versions in an epidemiological study.

We welcome this suggestion and have incorporated further contributions to the conclusions. As recommended in another point, we will also include a section on "applicability" which will allow focus more on this aspect.

Background:

5. Consider adding a small paragraph on different expected psychometric measures and in what way they can inform on DUFFS performances.

Thank you for the suggestion. We have included a paragraph on this issue in the introduction.

6. In the wording of the study objectives, it is unclear what is meant by “a standardised way” please consider specifying in what way.

A standardised assessment of the different versions of the DUFSS questionnaire was performed using the Evaluating the Measurement of Patient-Reported Outcomes (EMPRO) tool. To clarify this, the objective has been modified to read as follows: The objective of this study is to systematically review the available evidence on the psychometric and administration characteristics of the different versions of the DUFSS and to evaluate them through the standardised EMPRO tool.

Methods:

7. Line 95 – Please give more details on what articles were to be eligible. Are they accepted, published online, in paper, do they have to be peer reviewed? Are there any language restrictions? Are all language versions of the DUFFS to be included?

Thank you for giving us the opportunity to clarify this part of the methodology. We have tried to broaden the search as much as possible. Therefore, there are few restrictions, only the population under 18 years of age. There is no paper restriction, the older articles are on paper and have been digitised by scanning, so they are now available in digital format as well. We have also not restricted by language. But all the ones we have found were written in English or Spanish. The language in which they used the questionnaire to collect the data was the original language of each country, and that has not been considered excluding. This information has been included in the main document. In addition, following this proposal, we have modified Table 3 and have included a column detailing the language version of each article.

8. Line 98 – What I semant by “underage population”? Under the age of what?

Thank you for pointing out this term. We are referring to the population under 18 years of age. I am replacing the term in the manuscript.

9. Please consider moving lines 103-107 to the next section “search strategy”.

Thank you for this suggestion. The highlighted lines have been moved to the "Search strategy" section.

10. Discrepencies in what? Please reword to make clear if the selection process was done based on titles, than on abstracts than on full text and if discrepancies in choices for selection were treated during each step.

Thank you for allowing us to explain this point in more detail. 

We have included a detailed description of the process in the main manuscript. The paragraph reads as follows:

Initially, two reviewers screened the titles according to the inclusion criteria. Then, the same two reviewers did the same for the abstracts. Once the duplicates had been removed and based on the selection criteria, the eligibility of the full articles was assessed. Discrepancies that arose at each of the selection stages were resolved by discussion and consensus between the two researchers, and a third reviewer was consulted when consensus could not be reached between the two previous reviewers.

11. Line 116 – Please provide further details on what is meant by “main results”. It is useful to understand what psychometric measures were of interest and how they were extracted.

Thank you for this proposal. We have included more details in this section on the data extracted from each study in the "data items" section.

12. For compatibility between studies, were some measures recalculated based on the information made available?

The EMPRO tool is completed by expert evaluators in metric properties considering both information on results and on the quality of the methods applied in each study, so it was not necessary to recalculate any measure. 

13. Line 122 – Consider rewording the section heading to “EMPRO tool” and then “Experts”. Please provide further information about the EMPRO tool including its available psychometrics. What does this instrument measure and how valid is this measurement?

Thank you for this recommendation which will help for a better understanding.

We have made the indicated change of section.

14. How was study quality assessed for the included articles?

EMPRO contains several criteria aimed at assessing study quality in each of the following 5 attributes: conceptual and measurement model, reliability, validity, responsiveness and interpretability. Some of the EMPRO items assessing the results and methodological quality are: item 8-Linguistic equivalence; item 12-Cronbach’s alpha adequate; item 17-Reproducibility coefficients adequate; item 22-Prior hypothesis stated; and item 25-Adequacy of methods. Following the comment of the evaluator, we have considered it convenient to include in Table 2, where the EMPRO items are specified, information on the items that value results and those that value more the methodological quality of the studies.

The authors of the EMPRO (Valderas et al., 2008) defined the quality of a PRO measure as the ''degree of confidence that all possible biases have been minimised and that information about the process leading to its development and evaluation is clear and accessible''. The EMPRO combines 3 key aspects: (1) well-described and established attributes for assessment, (2) expert reviewers to conduct the assessment, and (3) scores that allow direct comparison between outcome measures. It is based on a comprehensive set of recommendations on the ideal attributes of PRO measures. The EMPRO is a valid and reliable tool that has proven useful for comparing the performance of generic PROs and disease-specific PROs, such as heart failure and shoulder disorders.

For this reason, we thought that the EMPRO items jointly assess both the metric properties and the methodological quality of the included studies and were therefore selected for this study.

Results:

15. Consider providing more detailed results of extracted psychometrics from the studies in the result section for Conceptual and measurement model, reliability, validity, responsiveness, and interpretability as reported in the discussion.

We are grateful for this recommendation, which helps to make the results easier to understand. Following your advice, we have rewritten the results along the lines of the discussion section. This makes the results section more focused and allows us to follow the rest of your recommendations.

15b. Adding a table with psychometric values for each version would help.

Table 4 contains the results of the EMPRO assessment according to each version of the DUFSS questionnaire. For clarification purposes, the name of the table has been changed in the table document and in the results section of the main document. The name assigned to the table is: Table 4. EMPRO results: psychometric values of each version of the DUFSS questionnaire.

Discussion:

16. Please consider synthesising the results in a paragraph, then to identify limitation of the actual study, and then discuss the results compared to other similar studies on similar instruments, then discuss the practical implications and then conclude.

We have followed your indications and the discussion has been reduced. The structure you have indicated has been followed and aspects of similar studies have been included throughout the discussion.

17. Lines 293-301, please provide deeper insights on cultural and language differences between versions of a questionnaire. To what extend can we assume we are measuring the same thing? What does the literature say about this and in what way could the results between different versions be explained by cultural differences rather than version differences? Do the different versions have similar psychometric values between English and Spanish (to be reported in the results section)?

Thank you for focusing on this important aspect. We agree that we should pay more attention to the description of these differences in the results and include a more detailed section in the discussion. Therefore, in addition to having included a detailed column on the language versions of the included questionnaires in table 3, the results and the discussion have been written with a focus on this aspect.

18. The main limitation is the reliance on the EMPRO tool. It is unclear whether a score can be affected only by good-bad psychometric performances or if it can be affected by the fact some psychometric measures were not made available by research. It is also unclear why each criteria has similar weight for the general score. In what way does that relate to the test value. A test that performed very badly in reliability will remain bad even if it performs well for other psychometric characteristics. Another limitation is combining psychometrics of tests run in different languages. Another limitation is that the version are compared only on psychometric performance but other advantages might exist on some versions over others such as time needed to answer questions, understandability, etc.

The data provided in the work of Valderas et al. suggest that the EMPRO instrument is valid. This tool demonstrated excellent reliability in terms of internal consistency (Cronbach's alpha = 0.95) and inter-rater concordance (intraclass correlation coefficient: 0.87-0.94).

The expected associations between EMPRO scores and the proposed variables were observed, supporting the construct validity of the tool. However, these relationships are consistent with the hypothesis that EMPRO scores depend on the quantity and quality of information provided for the assessment. Thus, psychometric measures that are not made available to the reader are not taken into account and therefore penalise the assessment. On the other hand, EMPRO offers the possibility to answer with "no information" or "not applicable". Items answered as ''no information'' are assigned a score of 1 (the lowest possible score) if at least 50 % of all items of an attribute were scored; b) items scored as ''not applicable'' (an option that is only available as a response for 5 items) were not taken into account in the scoring of the attributes. 

Another aspect that has been evaluated is the burden of administering the questionnaire. In this study, the two shorter versions were found to score better on this attribute.

Following this recommendation, we have synthesised the limitations of the study to the most relevant aspects of the evaluation through the EMPRO tool and have included the aspects indicated.

19. Lines 311-319; This does not seem to be a limitation as it does not seem to affect the results of this study. Misclassification of studies as been the DUFFS would be a limitation. What measures were taken to make sure that the retained article truly tested the DUFFS?

We agree with your point as it did not affect the results and did not lead to a misclassification of the DUFSS. In order to ensure that all validation articles on the DUFSS were located, we included the possible names it may acquire in the search strategy (this part is explained in the methodology); and once we had the search results, we went on to read the full text of any paper that might raise doubts, in order to ensure that it was the right questionnaire.

Although this is not a limitation for the results of our work, we think it is interesting to point out that the names and acronyms used for this questionnaire are different depending on each author, and that there are other different questionnaires with similar names that can be misleading.

20. Please consider adding a section on practical implication for research, education and public health.

Thank you for this suggestion, we think it is very appropriate to include this section in our manuscript. It will read as follows:

This paper can shed light on the study of social support as a PRO in different domains and help to unravel the current complexity that exists around this questionnaire. 

To know the different versions of the DUFSS questionnaire and providing relevant information about each one will allow researchers who wish to study this subject to choose the version that best suits their interests and to be aware of the evaluation of its quality. In addition, improving knowledge about this PRO will allow progress to be made and give greater strength to the work in the field of epidemiology and public health on person-centred care.

Furthermore, in the educational field, this study has two applications: on the one hand, to train specific tools in the study of PROs; on the other hand, it focuses on the importance of consulting the original sources and investigating the work that has been done previously on the research question.

Conclusion:

21. In what way can the 5-item version compete as it only was evaluated by two studies. How can authors justify this classification based on a score that could change in time depending of future available studies on other versions. How can we go beyond saying that with the actual available evidence, the DUFFS that shows the best performances is the 11-item version.

Following your previous indications, we have discussed this topic in the limitations section. leaving a related conclusion on the outcome in the conclusions section. The text we have included is as follows:

There are 4 versions of the EMPRO questionnaire with different numbers of items: 14, 11, 8 and 5 items. All of them have been validated in very specific populations and not in the general population. 

Among the 4 versions the DUFSS questionnaire, the 11-item version has been the most studied, especially in Spanish-speaking countries. This version scored higher than the others because it was the version with the largest number of studies and therefore more likely to address all the attributes taken into account by the EMPRO tool.

In order to be able to state with certainty that the 11-item version is more appropriate than the other versions, more studies are needed to evaluate each of the other versions. Although, a priori, we could prioritise its use over the other versions, it should be taken into account that this version should also be used with caution because there are attributes that have not been studied.

Figures:

22. Some text within the boxes have not been translated from Spanish.

Thank you for notifying us of this error. The untranslated texts have been corrected.

Tables and supplement data:

23. Extracted psychometric values for each version is not made available.

The results are not available for each of the items. The studies included in this work have been grouped by type of version based on the number of items obtained as a result of their validation of the DUFSS (14-item, 11-item, 8-item and 5-item version). Once each version has been grouped and studied through EMPRO, the results are shown in table 5. In order to clarify this aspect, the following paragraph has been included in the methodology:

The selected studies were grouped by version type according to the number of items that made up the version resulting from their study. The unit of analysis through EMPRO was each version type of the DUFSS. 

Another aspect is the aforementioned quality of the methods applied in each study. The EMPRO tool is completed by expert evaluators in metric properties who take into account both the outcome information and the methodological quality of the evaluated studies.

References:

Argimon Pallás, J. M. a. (2019). Métodos de investigación clínica y epidemiológica (S. L. U. Elsevier España, Ed.; Elsevier España, Vol. 5).

Ayala, A., Rodríguez-Blázquez, C., Frades-Payo, B., Forjaz, M. J., Martínez-Martín, P., Fernández-Mayoralas, G., Rojo-Pérez, F., Grupo Español de Investigación en Calidad de Vida y Envejecimiento, A., A., C., R.-B., B., F.-P., M.J., F., P., M.-M., G., F.-M., & F., R.-P. (2012). Psychometric properties of the Functional Social Support Questionnaire and the Loneliness Scale in non-institutionalized older adults in Spain. Gaceta Sanitaria, 26(4), 317–324. https://doi.org/10.1016/j.gaceta.2011.08.009

de la Revilla Ahumada, L., Bailón, E., de Dios Luna, J., Delgado, A., Prados, M. A., & Fleitas, L. (1991). Validation of a functional social support scale for use in the family doctor’s office. Atencion Primaria, 8(9), 688–692.

Services, H. (2006). Guidance for industry: Patient-reported outcome measures: Use in medical product development to support labeling claims: Draft guidance. Health and Quality of Life Outcomes, 4, 1–20. https://doi.org/10.1186/1477-7525-4-79

Valderas, J. M., Ferrer, M., Mendívil, J., Garin, O., Rajmil, L., Herdman, M., & Alonso, J. (2008). Development of EMPRO: A tool for the standardized assessment of patient-reported outcome measures. Value in Health, 11(4), 700–708. https://doi.org/10.1111/j.1524-4733.2007.00309.x

W.E. Broadhead, H Stephen Gehlbach, Frank V de Gruy, & Berton H Kaplan. (1988). The Duke-UNC Funtional Social Support Questionnaire. Measurement of Social Support in Family Medicine Patients. Medical Care, 26(7), 709–723.

---

## [Decision Letter · Decision Letter 1]

2 Aug 2023

PONE-D-22-28641R1Functional Social Support: a systematic review and standardized comparison of different versions of the DUFSS questionnaire using the EMPRO tool.PLOS ONE

Dear Dr. Lozano-Hernández,

Thank you for submitting your manuscript to PLOS ONE. After careful consideration, we feel that it has merit but does not fully meet PLOS ONE’s publication criteria as it currently stands. Therefore, we invite you to submit a revised version of the manuscript that addresses the points raised during the review process.

The reviewers have re-assessed your manuscript and are satisfied that you have addressed their comments. However, there are concerns that your submission does not yet fully comply with our data availability policy (https://journals.plos.org/plosone/s/data-availability). Authors must share the “minimal data set” for their submission. PLOS defines the minimal data set to consist of the data required to replicate all study findings reported in the article, as well as related metadata and methods. Additionally, PLOS requires that authors comply with field-specific standards for preparation, recording, and deposition of data when applicable. We would expect, for example, that sharing the completed data extraction form may be sufficient for another researcher to replicate your analysis. This should be labelled clearly for readers to understand and ideally in English. In addition, you may wish to review our competing interest form available at https://journals.plos.org/plosone/s/competing-interests to ensure that your competing interest statement is compliant.

We look forward to receiving your revised manuscript.

Kind regards,

Marianne Clemence

Staff Editor

PLOS ONE

Journal Requirements:

Reviewers' comments:

Reviewer's Responses to Questions

**Comments to the Author**

1. If the authors have adequately addressed your comments raised in a previous round of review and you feel that this manuscript is now acceptable for publication, you may indicate that here to bypass the “Comments to the Author” section, enter your conflict of interest statement in the “Confidential to Editor” section, and submit your "Accept" recommendation.

Reviewer #1: All comments have been addressed

Reviewer #2: (No Response)

2. Is the manuscript technically sound, and do the data support the conclusions?

Reviewer #1: Yes

Reviewer #2: Yes

3. Has the statistical analysis been performed appropriately and rigorously? 

Reviewer #1: N/A

Reviewer #2: N/A

4. Have the authors made all data underlying the findings in their manuscript fully available?

Reviewer #1: Yes

Reviewer #2: Yes

5. Is the manuscript presented in an intelligible fashion and written in standard English?

Reviewer #1: Yes

Reviewer #2: Yes

6. Review Comments to the Author

Reviewer #1: (No Response)

Reviewer #2: Dear authors,

The article is relevant, complete and valuable to both research community in social epidemiology and for accounting for social factors in any clinical study. The changes brought to the “Practical application” section seems to hit spot on to make readers understand why this study is so important. Table 5 provides a clear transparent view of the available information and the EMPRO results. I also particularly appreciated the detailed response to all our comments. This has been thoroughly done and very well argued making the review process so much easier at this stage.

--- Minor suggestions ---

1. Competing interests - Authors might consider relying on a statement of interest rather than a statement of conflict of interest. Indeed, it is up to the reader to judge whether or not whatever is stated constitutes a conflict. For this reason, it is often recommended to state education interests and academic positions in the field, grant obtention, payed fees for consultations, participation to interest groups such as association, foundations, etc, contribution as scientific advisor for working groups, think tanks, etc., clinical activities in relation to the topic, etc. Readers slowly tend to find it suspicious if no interest are declared. Siting many interests that could constitute a better understanding of the underlying cognitive biases helps readers feel that the work is trustworthy.

2. Data availability: It could be seen as misleading to say all data is made available without restriction. Indeed, the working documents are not made available. It would be interesting to have the individual study assessments made using EMPRO made available. If you want to do this, consider depositing them on a repository such as on Zenodo.org

3. Line 342, the citation to the reference for Broadhead et al. still needs to be added.

7. PLOS authors have the option to publish the peer review history of their article (what does this mean?). If published, this will include your full peer review and any attached files.

Reviewer #1: No

Reviewer #2: **Yes: **Paul Vaucher

---

## [Author Response · Author response to Decision Letter 1]

16 Aug 2023

Dear reviewers:

Thank you for the thorough review. We appreciate the improvements made to our manuscript.

In addition to including the contributions requested by the reviewers, considerations made by the editor regarding format and style have been reviewed and incorporated into the manuscript.

Each of the requested responses is detailed below.

Reviewer #2: 

Dear authors,

The article is relevant, complete and valuable to both research community in social epidemiology and for accounting for social factors in any clinical study. The changes brought to the “Practical application” section seems to hit spot on to make readers understand why this study is so important. Table 5 provides a clear transparent view of the available information and the EMPRO results. I also particularly appreciated the detailed response to all our comments. This has been thoroughly done and very well argued making the review process so much easier at this stage.

--- Minor suggestions ---

1. Competing interests - Authors might consider relying on a statement of interest rather than a statement of conflict of interest. Indeed, it is up to the reader to judge whether or not whatever is stated constitutes a conflict. For this reason, it is often recommended to state education interests and academic positions in the field, grant obtention, payed fees for consultations, participation to interest groups such as association, foundations, etc, contribution as scientific advisor for working groups, think tanks, etc., clinical activities in relation to the topic, etc. Readers slowly tend to find it suspicious if no interest are declared. Siting many interests that could constitute a better understanding of the underlying cognitive biases helps readers feel that the work is trustworthy.

As recommended, a declaration of interests has been included.

2. Data availability: It could be seen as misleading to say all data is made available without restriction. Indeed, the working documents are not made available. It would be interesting to have the individual study assessments made using EMPRO made available. If you want to do this, consider depositing them on a repository such as on Zenodo.org

Following your recommendation, the dataset corresponding to the evaluations of the different versions of the DUFSS questionnaire using the EMPRO tool has been published in Zenodo. 

The publication is under DOI:

https://doi.org/10.5281/zenodo.8211219

3. Line 342, the citation to the reference for Broadhead et al. still needs to be added.

The requested reference has been included. In addition, the reference list has been revised and updated.

---

## [Decision Letter · Decision Letter 2]

4 Sep 2023

Functional Social Support: a systematic review and standardized comparison of different versions of the DUFSS questionnaire using the EMPRO tool.

PONE-D-22-28641R2

Dear Dr. Cristina M Lozano-Hernández

We’re pleased to inform you that your manuscript has been judged scientifically suitable for publication and will be formally accepted for publication once it meets all outstanding technical requirements.

Kind regards,

Victor Manuel Mendoza-Nuñez, PhD

Academic Editor

PLOS ONE

---

## [Editor Report · Acceptance letter]

7 Sep 2023

PONE-D-22-28641R2 

Functional Social Support: a systematic review and standardized comparison of different versions of the DUFSS questionnaire using the EMPRO tool. 

Dear Dr. Lozano-Hernández:

I'm pleased to inform you that your manuscript has been deemed suitable for publication in PLOS ONE. Congratulations! Your manuscript is now with our production department. 

Kind regards, 

on behalf of

Dr. Victor Manuel Mendoza-Nuñez 

Academic Editor

PLOS ONE